# Climatic Factors Determine the Distribution Patterns of Leaf Nutrient Traits at Large Scales

**DOI:** 10.3390/plants11162171

**Published:** 2022-08-21

**Authors:** Xianxian Wang, Jiangfeng Wang, Liuyang Zhang, Chengyu Lv, Longlong Liu, Huixin Zhao, Jie Gao

**Affiliations:** 1College of Life Sciences, Xinjiang Normal University, Urumqi 830054, China; 2Hubei Forestry Investigation and Planning Institute, No. 4, Zhuodaoquan South Road, Hongshan District, Wuhan 430070, China; 3Taiyuan Ecology and Environment Monitoring & Science Research Center, Taiyuan 030002, China; 4East China Survey and Planning Institute, National Forestry and Grassland Administration, Hangzhou 310019, China; 5Institute of Ecology and Key Laboratory of Earth Surface Processes of Ministry of Education, College of Urban and Environmental Sciences, Peking University, Beijing 100871, China

**Keywords:** leaf nutrients, functional traits, life forms, climate change, soil nutrients

## Abstract

Leaf nutrient content and its stoichiometric relationships (N/P ratio) are essential for photosynthesis and plant growth and development. Previous studies on leaf nutrient-related functional traits have mainly focused on the species level and regional scale, but fewer studies have investigated the distribution patterns of the leaf N and P contents (LN, LP) and N/P ratios (N/P) in communities and their controlling factors at a large scale; therefore, we used LN, LP, and N/P data at 69 sites from 818 forests in China. The results showed significant differences (*p* < 0.05) in the LN, LP, and N/P at different life forms (tree, shrub, and herb). Neither LN, LP, nor N/P ratios showed significant patterns of latitudinal variation. With the increase in temperature and rainfall, the LN, LP, and leaf nutrient contents increased significantly (*p* < 0.001). Across life forms, LN at different life forms varied significantly and was positively correlated with soil P content (*p* < 0.001). The explanatory degree of climatic factors in shaping the spatial variation patterns of LN and N/P was higher than that of the soil nutrient factors, and the spatial variation patterns of the leaf nutrient traits of different life forms were shaped by the synergistic effects of climatic factors and soil nutrient factors.

## 1. Introduction

The functional traits of plants reflect their survival strategies in response to environmental changes [1,2]. Nitrogen (N) and phosphorus (P) are important components of the basic structure of plant cells and their levels. The stoichiometric relationships between N and P drive photosynthesis, plant growth, reproduction, and other life processes [3]. Leaf nitrogen (LN), phosphorus (LP), and leaf N/P ratios (N/P) are key plant traits that influence the productivity of forest communities and regulate carbon cycling [4]. Numerous studies have found that climatic factors, soil nutrient factors, and community genealogical structure influence the plant functional traits by affecting plant metabolism [5]. Exploring the distribution patterns of functional traits related to leaf nutrients on a large scale has important ecological significance for quantifying the impact of environmental changes on plant functional traits [6]. However, the key drivers that determine the distribution patterns of these key functional traits remain elusive.

Climate factors influence the functional traits of plant leaves by controlling plant metabolism, growth, and development processes [7]. A global data-based study found that climate change significantly affected the leaf nutrient content [8]. With increasing temperature, the LN decreases significantly and LP increases significantly [9]. Higher LN enhances photosynthesis under low-temperature stress, and when temperatures are higher, plants coordinate metabolic processes by reducing the LN and increasing LP [10]. Other studies have found that the N/P of plant leaves increased significantly with increasing temperature [11]. Numerous previous studies have shown that temperature is a key factor in determining changes in leaf nutrient traits, especially the mean annual temperature (MAT) [4,12,13].

Plant leaf nutrients are also influenced by precipitation factors [14]. Precipitation factors can alter soil nutrient availability, and when rainfall is low and soils are subject to drought stress, plants respond to water stress by changing functional traits [15]. Temperature and precipitation factors jointly determine the nutrient changes in plant leaves [16]. In specific forest communities, the key climatic factors that determine the leaf nutrients in different species remain elusive due to differences in biological adaptations and plant nutrient uptake strategies. Therefore, it is particularly important to quantify the effects of climatic factors in the leaves of different plant forms at a macroscopic scale.

Not only do climatic factors have an important impact on the nutrient profiles of plant leaves, but soil nutrient factors also play a key role in shaping the spatiotemporal pattern of leaf nutrient traits [17]. Climate affects the plant nutrient profiles by influencing soil nutrient redistribution [18]. Plant vessels absorb inorganic salts and water from the soil and transport them to the leaves, so the soil is the main source of nitrogen and phosphorus for plant leaves [19]. The nutrient profiles of leaves are limited by the nutrient availability of the soil [20]. Some studies have found that the N/P of Chinese plant leaves is significantly lower than the global average due to the low effective phosphorus content of Chinese soils [21]. It has been shown that the N and P content of plant leaves increases with increasing soil N content and soil pH [22], and that soil nutrient factors directly affect the nutrient profiles of leaves. 

Forest communities are divided into tree, shrub, and herb levels according to species composition, structure, and production, with each plant having its life form, and vertical differentiation provides a good indication of the community’s adaptation to environmental conditions [23]. The nutrient profiles of leaves vary significantly among different life forms due to differences in their survival strategies [24]. Comparisons of the leaf nutrient traits of different plant forms are mostly based on the species scale or local scale, and few studies have been conducted on a larger spatial scale based on the community scale. Community trait variation is more effective than species trait variation in predicting the plant response to environmental change [25]. However, due to interspecific competition and intraspecific struggle in a given forest community, there is a great theoretical risk in predicting the functional traits of individual plants [26]. Therefore, the effects of environmental factors on community leaf nutrient traits at a macro scale can reduce the impact of certain local deterministic processes (e.g., competition, mutualism). 

Based on field survey data from 89 sites in China, an attempt was made to identify the key drivers influencing the leaf nutrient traits at different life forms in forest communities at the macro scale. To explain this, the following hypotheses were made: (1) there are significant differences in leaf nutrient traits at different life forms at the macro scale; and (2) climatic factors are the dominant environmental factors affecting leaf nutrient traits at different life forms of the community, while soil nutrient factors also play a coordinating role that cannot be ignored.

## 2. Results

### 2.1. Variability in LN, LP, and N/P of Different Life Forms and Distribution Patterns in China

Leaf nutrient traits showed significant geographical variation at different life forms, with overall higher LN and LP in northeastern China and relatively higher LP content in northwestern China (Figure 1). However, neither LN, LP, nor N/P (Figure 2) showed a significant latitudinal pattern. The LN, LP, and N/P of the herb levels showed significant differences (*p* < 0.05). The leaf nutrient traits of the arbor layer and the herb layer were significantly different (*p* < 0.001), and the herb layer had relatively higher LN and lower LP. Leaf nutrient traits were generally significantly different between the shrub and herb levels (*p* < 0.05), but LN was not significant in the herb and shrub levels (*p* > 0.05).

### 2.2. Correlations between Climatic Factors and LN, LP, and N/P at Different Life Forms

The LN and LP of the tree, shrub, or herb levels increased significantly with increasing MAT and mean annual precipitation (MAP) (Figure 3a,d and Figure 4a,d) and decreased significantly with increasing annual sunlight duration (ASD) (Figure 3e,f and Figure 4e). Whereas LP in both the tree and shrub levels increased significantly with increasing mean coldest monthly temperature (MCMT) and mean warmest monthly temperature (MWMT), LN in the herb level decreased significantly. The ASD and mean annual evaporation (MAE) had better predictive power for LN in the tree level compared to other climatic factors (*R^2^* = 0.20, *p* < 0.001; *R^2^* = 0.16, *p* < 0.001) and better predictive power for LP in the tree layer (*R^2^* = 0.46, *p* < 0.001; *R^2^* = 0.50, *p* < 0.001; *R^2^* = 0.37, *p* < 0.001) were MWMT, MAP, and ASD. 

The N/P of the herb level increased with the increase in MCMT and MWMT (Figure 5b,c), and decreased with the increase in MAT, MAP, ASD, and MAE (Figure 5a,d–f). MAT, MCMT, and MWMT (Figure 5a–c) were significantly correlated with the N/P of the shrub layer (*p* < 0.001). MAT had the best fitting effect on the N/P of the herbal layer (Figure 5b; *R^2^* = 0.59, *p* < 0.001).

### 2.3. Effect of Soil Factors on the Relationship between LN, LP, and Leaf N/P at Different Life Forms

The LN of the different life forms was significantly and positively correlated with the soil P (Figure 6b), while the trends of LP and N/P varied with the soil nutrient factors. The LN of the three life forms increased significantly with increasing soil N and P (Figure 6). Soil pH (Figure 7c) all showed a significant positive correlation (*p* < 0.001) with LP for the different life forms, whereas LN decreased significantly with increasing soil pH (Figure 6). The LP of the shrub levels increased significantly with the increasing soil N and P. The trend between the herb and shrub levels was reversed. The N/P in the herb level was significantly positively correlated with soil N (*R^2^* = 0.30, *p* < 0.001) (Figure 8a). Soil pH (Figure 7c) was the best predictor of LP in the herb and tree levels (*R^2^* = 0.25, *p* < 0.001; *R^2^* = 0.29, *p* < 0.001) (Figure 7c), and the best predictor of N/P in the tree level (Figure 8c) (*R^2^* = 0.29, *p* < 0.001) (Figure 8c). In contrast, the best prediction of N/P for the shrub level was for soil P (*R^2^* = 0.22, *p* < 0.001; Figure 8b).

### 2.4. Climatic and Soil Factors Dominate Changes in the Functional Traits of Different Communities

We analyzed the effects of environmental factors on the leaf LN, LP, and N/P at different life forms based on a generalized additive model with non−metric multidimensional scaling (NMDS) ranking. Overall, the leaf N/P showed strong environmental plasticity, and environmental factors generally explained the spatial variation in leaf N/P more than LN and LP at different life forms (Figure 9, Figure 10 and Figure 11). Climate factors played a greater role than the soil nutrient factors in shaping the spatial variation in the leaf LN (Figure 9d−f; de = 42.2%, 41%, 22.8%) and LP (Figure 10d−f; de = 17.2%, 17.1%, 50.6%). 

## 3. Discussion

The leaf nutrient traits (LN, LP, and N/P) of different plant forms showed significant geographical differences [27], and some studies have found that the LN of shrub levels is significantly higher than that of the tree and herb levels in northwestern China, while the LP of shrub levels is significantly lower than that of herb levels [28], and that the LP and N/P of shrubs are higher than those of herbs in the desertification zone of southwest Hunan [29]. Within a given community, differences in the ecological niches of species and differences in resource use patterns can lead to differences in the leaf nutrient traits between species [17]. Leaf nutrient traits are widely used to quantify plant adaptations to the environment, and even to predict ecosystem function [6,30]. Thus, the differences in LN and LP of different life forms of plants result from differences in plant ecological niches. Compared to herb plants, woody plants have lower LP, which is aligned with their low growth rate strategy [31]. Quantifying the geographical distribution patterns of leaf nutrient traits is important for our scientific assessment of forest development dynamics [24].

Temperature can have a significant effect on the LN, LP, and N/P [32], and the temperature−biogeochemical hypothesis also suggests that LN and LP increase monotonically with temperature on a global scale [33]. It has been found that LN and LP are strongly correlated with effective soil N and P content [32], and that climatic factors can act directly on soil nutrients [25]. Soil enzymes are beneficial to maintain soil fertility, and increasing the soil temperature and moisture can promote the activity of soil enzymes and significantly improve the decomposition efficiency of soil organic matter [34]. Low temperatures not only have an inhibitory effect on organic matter decomposition and mineralization, but also limit microbial activity and affect the decomposition of apoplastic matter, thus reducing the availability of soil N and P [33]. Conversely, as the temperature increases, soil microbial respiration is enhanced, the efficiency of organic matter mineralization and decomposition increases, and the effective soil N, and P content rises [35]. Therefore, MAT is a key limiting factor for changes in leaf nutrient traits [12,13]. It has also been found that ASD significantly affects LN and LP [36]. Plant N and P elements are involved in the light reaction process and have an impact on photosynthesis, which is enhanced with increasing light duration and LN and LP are heavily utilized [37]. The nutrient content of plant leaves gradually decreased with increasing light time. Our results demonstrate the important influence of temperature on the nutrients of different life forms of leaves.

Precipitation is also one of the main factors of nutrient traits in plant leaves [14]. Previous studies have found that in water−limited ecosystems, plant N and P uptake of elements is mainly limited by water [38]. On one hand, most soil water comes from rainfall, and soil water content can directly affect the activity of soil enzymes, thereby affecting the release of soil nutrients [34]. On the other hand, rainfall promotes plant N uptake by promoting soil element availability, and plant N uptake is positively correlated with soil N availability [39]. In addition, rainfall had a significant effect on the soil microbial respiration and accelerated the leaching and transformation of P, which contributed to the transformation of soil P and thus increased the effective soil P content. With increased precipitation, the effectiveness of the soil nutrient factors increased, and plants were able to take up more N and P, significantly increasing the LN and LP [40]. Thus, precipitation is an important effect on the spatial and temporal distribution patterns of plant nutrient traits.

Soil N and P are basic nutrients for plant growth and are closely related to plant leaf nutrient, and most of the LN and LP of plants originate from soil [9]. The availability of soil N and P elements directly determines the growth and development process of plants, and available soil nutrients are positively correlated with leaf nutrients [18]. It was found that fertilizing the soil with N and P was beneficial to maintaining and improving the activity of soil enzymes and releasing more micronutrients and nutrients [41]. The higher the effective soil N and P content, the higher the plant root N and P uptake efficiency, and the higher the LN and N/P [22]. Soil microbial activity is positively correlated with soil pH, especially the activity of soil microbial enzymes related to the breakdown of soil nutrients. As the pH increases, soil microbial metabolic activity decreases, inhibiting the soil N mineralization and plant uptake of soil N, and LN is subsequently reduced [42]. However, it was also found that soil microbial biomass increased with the increasing soil pH, effectively driving organic P mineralization and the dissolution of fixed P [43], and LP in the arboreal and herbaceous layers increased. Thus, soil nutrient factors significantly influenced the changes in the leaf nutrient traits.

At the macro scale, climate factors have a stronger ability to shape leaf traits than soil factors [44]. Studies have shown that plant leaves are more sensitive to changes in climate and that plants can respond to changing climate by changing their traits [1]. Soil nutrient elements are closely related to leaf nutrient elements and provide many nutrients for plant growth and development, but the availability of soil nutrients is limited by temperature and precipitation [33]. Thus, climatic factors play a stronger role in shaping leaf nutrient distribution patterns than soil factors [45], but the influence of soil nutrient factors on leaf traits is not negligible.

The research area of this thesis included most of areas, which together with the relatively large amount of data ensures the plausibility of our experimental results. In addition, the results of this study are presented using a variety of data analysis methods. Overall, our results quantify the relative contribution of different environmental factors in shaping the functional traits of leaf nutrients and are important for a better understanding of the impact of global climate change on plant physiology [10]. In future studies, more biological factors such as community biodiversity and stand density need to be further explored for their effects on the distribution patterns of community leaf nutrients.

## 4. Materials and Methods

### 4.1. Study Area and Sample Data

The relatively wide distribution of forests in China facilitates studies on large scales. Data from 818 forest at 89 sites plots surveyed between 2005 and 2020 were used to investigate the spatial distribution and driving factors of LN, LP, and N/P at different life forms (tree levers, shrub levels, herb levels) of forest communities (Appendix A). The study sites ranged from 19.1° to 53.5° N and 79.7° to 129.3° E.

### 4.2. Functional Data

Community nutrient profiles can better reflect the adaptability of local vegetation to different environments than individual nutrient profiles [46]. Data for the selected nutrient profiles in this study include LN, LP, and N/P.

In this experiment, LN was determined using the national standard method, the Kjeldahl method [47], LP was determined using the vanadium−molybdenum yellow absorbance method [48], and N/P was equal to LN/LP. 

Theoretical risk in predicting functional traits in individual plants is due to intraspecific and interspecific struggles [49]. Community−weighted mean traits (*CWM*_i_) represents the forest mean trait values.
$$CWM_{i}=\frac{\sum_{i}^{n}D_{i}\times Trait}{\sum_{i}^{n}D_{i}}$$
where *CWM*_i_ represents the community−weighted functional trait identity value and *D*_i_ represents the abundance of dominant tree species. *Trait*_i_ represents the selected functional trait [50]. 

### 4.3. Environmental Data

A study found that changes in the functional traits of building blocks were related to climate change [51]. The mean annual temperature (MAT), mean coldest monthly temperature (MCMT), mean warmest monthly temperature (MWMT), and mean annual precipitation (MAP) were extracted from the WorldClim global climate layer at a spatial resolution of 1 km. The mean annual evaporation (MAE) was taken from the Climate Data Center of the China Meteorological Administration (http://data.cma.cn/site/index.html) (accessed on 1 April 2021), sunlight. The annual sunlight duration (ASD), a key factor in photosynthesis [52], is from the Climate Data Center of the China Meteorological Administration (http://data.cma.cn/site/index.html) (accessed on 1 April 2021). Soil pH, soil N (http://www.csdn.store) (accessed on 1 April 2021), and soil P (https://www.osgeo.cn/data/wc137) (accessed on 1 April 2021) in the top 30 cm of soil were extracted from a 250 m resolution grid. 

### 4.4. Data Analysis

Data analysis of the LN, LP, and N/P data from our study conformed to a normal distribution (Appendix A). A significant difference test at the 0.05 significance level was used to test for significant differences in LN, LP, and N/P between the tree, shrub, and herb levels (Figure 2). Significant differences were analyzed using the R package agricolae (version 4.1.0, R Core Team, 2020). 

The extent to which environmental factors explain LN, LP, and N/P was investigated using a linear regression model in the R package agricolae (version 4.1.0, R Core Team, 2020), where *R^2^* represents how well the model fits the variables studied. 

The generalized additive model (GAM) is used to test the effects of various environmental factors on the functional traits of leaves, with data for the LN, LP, and N/P consisting of parametric and non−parametric components to reduce the model risk associated with linear models [49]. First, the GAM method was used to select the key influencing factors based on a significant difference test at the 0.05 level of significance. Then, a GAM model was constructed to measure the relationship between the environmental factors and LN, LP, and N/P and used non−metric multidimensional scaling analysis (NMDS) to reflect the results for GAM [53].
$$g[E(Y|X)]=\underset{i}{\sum }\beta iXi+\underset{j}{\sum }fi\left(Xi\right)+\varepsilon$$
where g(•) represents the connection function, the form depends on the specific form, which can be interpreted as the Y-variable distribution. Є is the random error term, which can be interpreted as the variable connection with the normal distribution function name identity, and the connection function has the form g(u) = u, u = E(Y|X), E(є|X) = 0. X_i_ is the explanatory variable that strictly follows the parametric form, β_i_ is the corresponding parameter, and f_j_(X_j_) is the corresponding explanatory variable that follows the nonparametric form of the smoothing function. In our study, the spline smoothing function S(•) was selected for fitting, thin−plate spline smoothing was selected for function fitting between different nodes, and each smoothing function S(•) was estimated using penalized least squares [50]. 

## 5. Conclusions

This study used data at 89 sites from 818 forest plots across China over 15 years from 2005 to 2020 to verify the relative roles of climatic and soil factors in shaping LN, LP, and N/P at different life forms. Our results confirm that LN, LP, and N/P differ significantly between life forms and that climatic and soil factors in the community habitat jointly influence the distribution patterns of the N/P nutrient profiles, with MAT being the dominant factor influencing LN, LP, and N/P at different life forms. Climatic factors also indirectly influence the leaf nutrient traits by affecting soil nutrients, but climatic factors are more influential than soil factors in shaping the distribution patterns of LN, LP, and N/P. This study has important implications for our understanding of the distribution patterns of the plant N and P nutrient profiles at different life forms in the context of climate change and ecosystem modeling.

## Figures and Tables

**Figure 1 plants-11-02171-f001:**
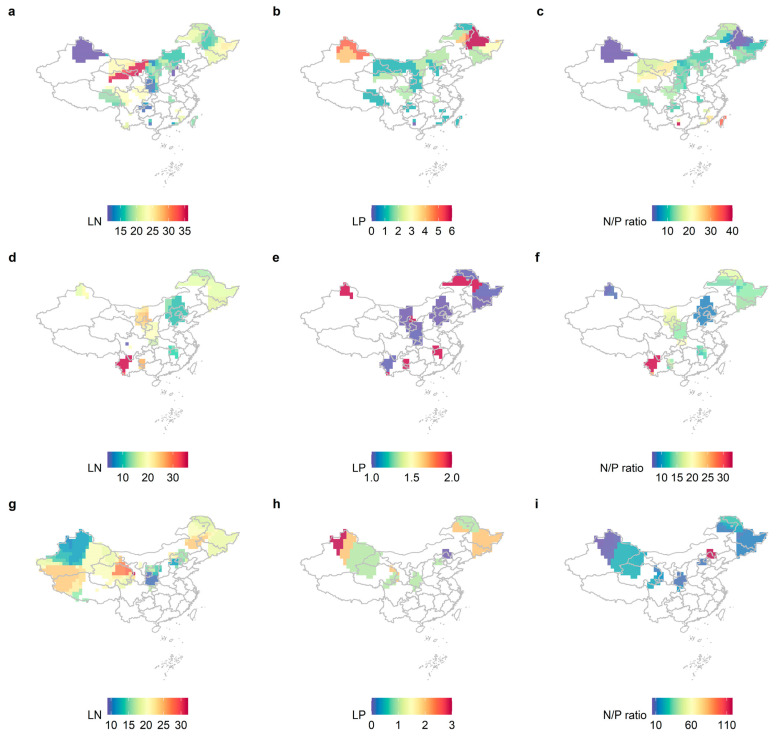
The distribution patterns of LN, LP, and N/P at different life forms in China with a spatial resolution of 1 × 1 km were studied by kernel density estimation. (**a**) LN of the tree levels; (**b**) LP of the tree levels; (**c**) N/P of the tree levels; (**d**) LN of the shrub levels; (**e**) LP of the shrub levels; (**f**) leaf N/P of the shrub levels; (**g**) LN of the herb levels; (**h**) herb levels LP; (**i**) herb levels N/P.

**Figure 2 plants-11-02171-f002:**
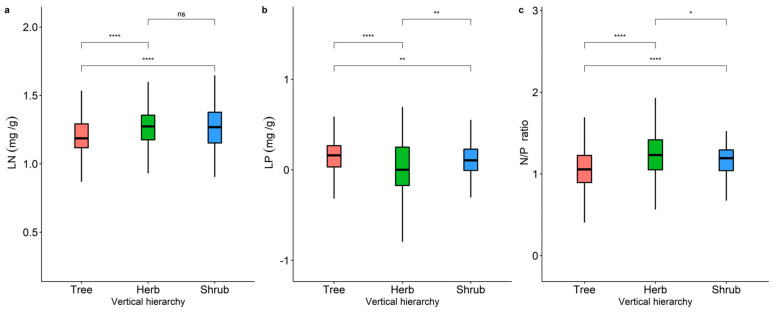
A comparison of the differences in the LN (**a**), LP (**b**), and N/P (**c**) at different life forms. (**a**) Variability of LN among the trees, shrubs, and herbs; (**b**) variability of LP among the trees, shrubs, and herbs; (**c**) variability of N/P among the trees, shrubs, and herbs. Tree represents the tree levels, Shrub represents the shrub levels, and Herb represents the herb levels. Levels are grouped where ns represents non−significant (*p* > 0.05) at the 0.05 level, * represents *p* < 0.05, ** represents *p* < 0.01, **** represents *p* < 0.0001.

**Figure 3 plants-11-02171-f003:**
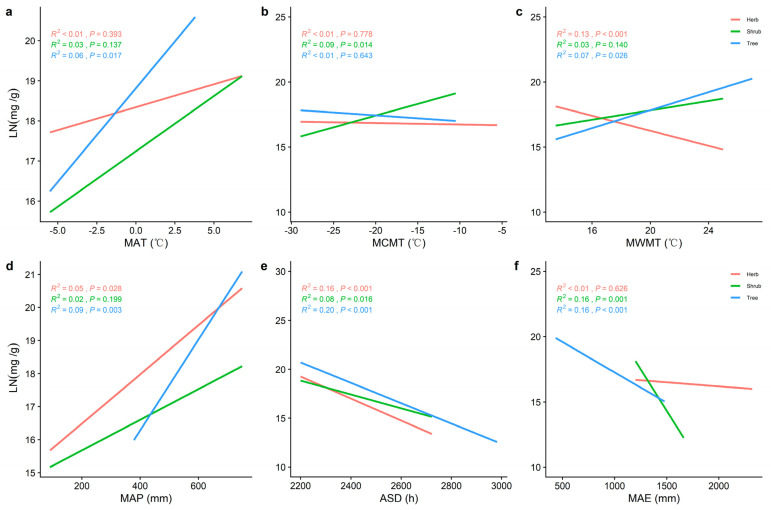
The general linear correlation analysis of climate factors with LN at different life forms. (**a**) General linear relationship between the MAT and LN of plants in the trees, shrubs, and herbs. (**b**) General linear relationship between the MCMT and LN of plants in the trees, shrubs, and herbs. (**c**) General linear relationship between the MWMT and LN of plants in the trees, shrubs, and herbs (**d**) General linear relationship between the MAP and LN of plants in the trees, shrubs, and herbs (**e**) General linear relationship between the ASD and LN of plants in the trees, shrubs, and herbs (**f**) General linear relationship between the MAE and LN of plants in the trees, shrubs, and herbs. MAT represents the mean annual temperature, MCMT represents the mean coldest monthly temperature, MWMT represents the mean warmest monthly temperature, MAP represents the mean annual precipitation, ASD represents the annual sunlight duration, and MAE represents the mean annual evaporation. The red line represents the tree level, the green line is the shrub level, and the blue line is the herb level. *R^2^* represents how well the model fits the variables studied and the *p*-value represents the significance level.

**Figure 4 plants-11-02171-f004:**
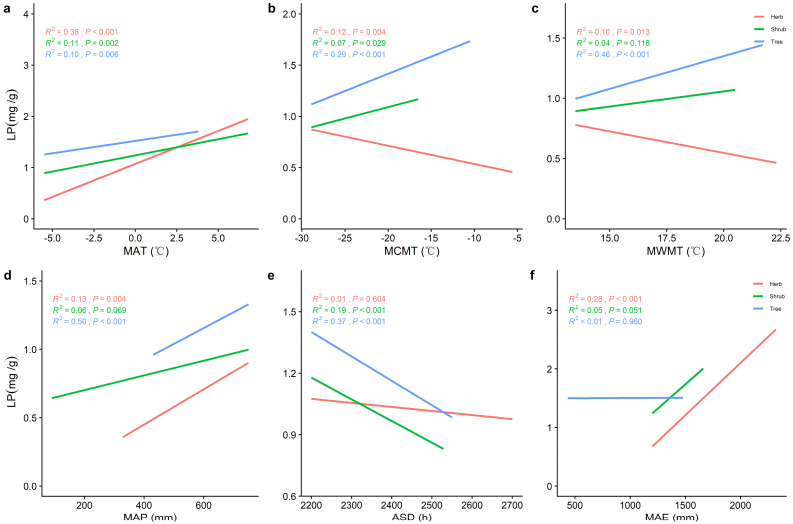
The general linear correlation analysis of climate factors with LP at different life forms. (**a**) General linear relationship between the MAT and LP of plants in the trees, shrubs, and herbs. (**b**) General linear relationship between the MCMT and LP of plants in the trees, shrubs, and herbs. (**c**) General linear relationship between the MWMT and LP of plants in the trees, shrubs, and herbs. (**d**) General linear relationship between the MAP and LP of plants in the trees, shrubs, and herbs. (**e**) General linear relationship between the ASD and LP of plants in the trees, shrubs, and herbs. (**f**) General linear relationship between the MAE and LP of plants in the trees, shrubs, and herbs. MAT represents the mean annual temperature, MCMT represents the mean coldest monthly temperature, MWMT represents the mean warmest monthly temperature, MAP represents the mean annual precipitation, ASD represents the annual sunlight duration, and MAE represents the mean annual evaporation. The red line represents the tree level, the green line is the shrub level, and the blue line is the herb level. *R^2^* represents how well the model fits the variables studied and the *p*-value represents the significance level.

**Figure 5 plants-11-02171-f005:**
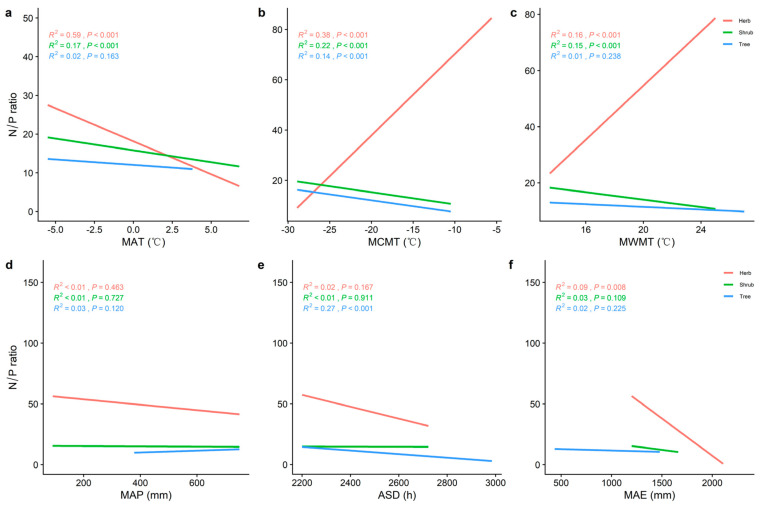
The general linear correlation analysis of climate factors with N/P at different life forms. (**a**) General linear relationship between the MAT and N/P of plants in the trees, shrubs, and herbs. (**b**) General linear relationship between the MCMT and N/P of plants in the trees, shrubs, and herbs. (**c**) General linear relationship between the MWMT and N/P of plants in the trees, shrubs, and herbs. (**d**) General linear relationship between the MAP and N/P of plants in the trees, shrubs, and herbs. (**e**) General linear relationship between the ASD and N/P of plants in the trees, shrubs, and herbs. (**f**): General linear relationship between the MAE and N/P of plants in the trees, shrubs, and herbs. MAT represents the mean annual temperature, MCMT represents the mean coldest monthly temperature, MWMT represents the mean warmest monthly temperature, MAP represents the mean annual precipitation, ASD represents the annual sunlight duration, and MAE represents the mean annual evaporation. The red line represents the tree level, the green line is the shrub level, and the blue line is the herb level. *R^2^* represents how well the model fits the variables studied and the *p*-value represents the significance level.

**Figure 6 plants-11-02171-f006:**
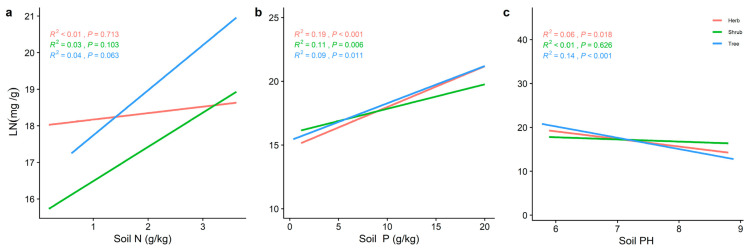
The general linear analysis of soil factors with different life forms of LN. (**a**) General linear relationship between the soil N and LN of plants in the trees, shrubs, and herbs. (**b**) General linear relationship between the soil P and LN of plants in the trees, shrubs, and herbs. (**c**) General linear relationship between the soil pH and LN of plants in the trees, shrubs, and herbs. The red line represents the tree level, the green line is the shrub level, and the blue line is the herb level. *R^2^* represents how well the model fits the variables studied and *p*-value represents the level of significance.

**Figure 7 plants-11-02171-f007:**
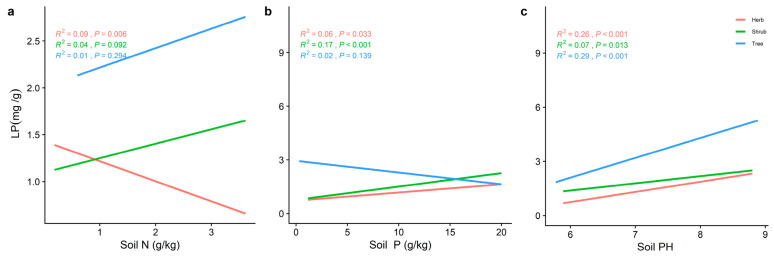
The general linear analysis of soil factors with different life forms of LP. (**a**) General linear relationship between the soil N and LP of plants in the trees, shrubs, and herbs (**b**) General linear relationship between the soil P and LP of plants in the trees, shrubs, and herbs. (**c**) General linear relationship between the soil pH and LP of plants in the trees, shrubs, and herbs. The red line represents the tree level, the green line is the shrub level, and the blue line is the herb level. *R^2^* represents how well the model fits the variables studied and the *p*-value represents the level of significance.

**Figure 8 plants-11-02171-f008:**
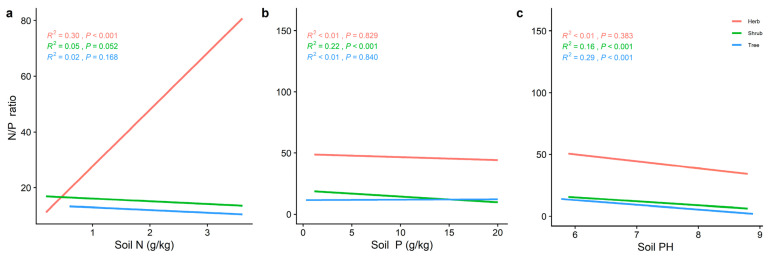
The general linear analysis of soil factors with different life forms of N/P. (**a**) General linear relationship between the soil N and N/P of plants in the trees, shrubs, and herbs (**b**) General linear relationship between the soil P and N/P of plants in the trees, shrubs, and herbs. (**c**) General linear relationship between the soil pH and N/P of plants in the trees, shrubs, and herbs. The red line represents the tree level, the green line is the shrub level, and the blue line is the herb level. *R^2^* represents how well the model fits the variables studied and the *p*-value represents the level of significance.

**Figure 9 plants-11-02171-f009:**
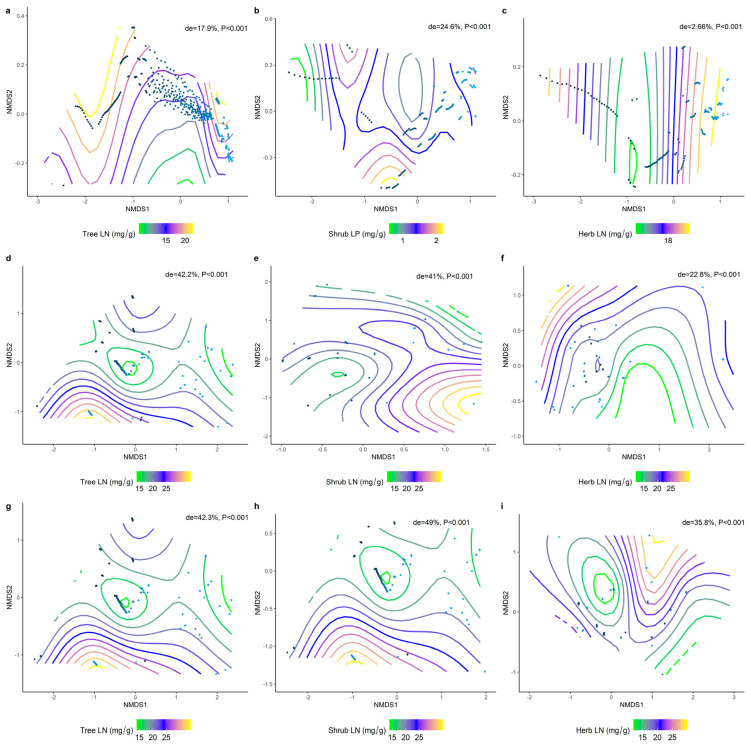
The NMDS ranking of climatic and soil factors with different life forms of LN. Value of de represents the deviation explained by the corresponding model. (**a**) NMDS ranking of soil factors with tree levels LN; (**b**) NMDS ranking of soil factors with shrub levels LN; (**c**) NMDS ranking of soil factors with herb levels LN; (**d**) NMDS ranking of climatic factors with tree levels LN; (**e**) NMDS ranking of climatic factors with shrub levels LN; (**f**) NMDS ranking of climate factors and herb levels LN; (**g**) NMDS ranking of the sum of soil factors and climate factors and tree levels LN; (**h**) NMDS ranking of the sum of soil factors and climate factors and shrub levels LN; (**i**) NMDS ranking of the sum of soil factors and climate factors and herb levels LN. Trait stacking indicates that abiotic factors, indicated by points on the NMD, are associated with higher or lower trait values, consistent with a colored trait gradient. Note that if the relationship between the LN and abiotic factors is linear, the gradient splines will be parallel. Nonlinear relationships between LN and abiotic factors are represented by curve splines.

**Figure 10 plants-11-02171-f010:**
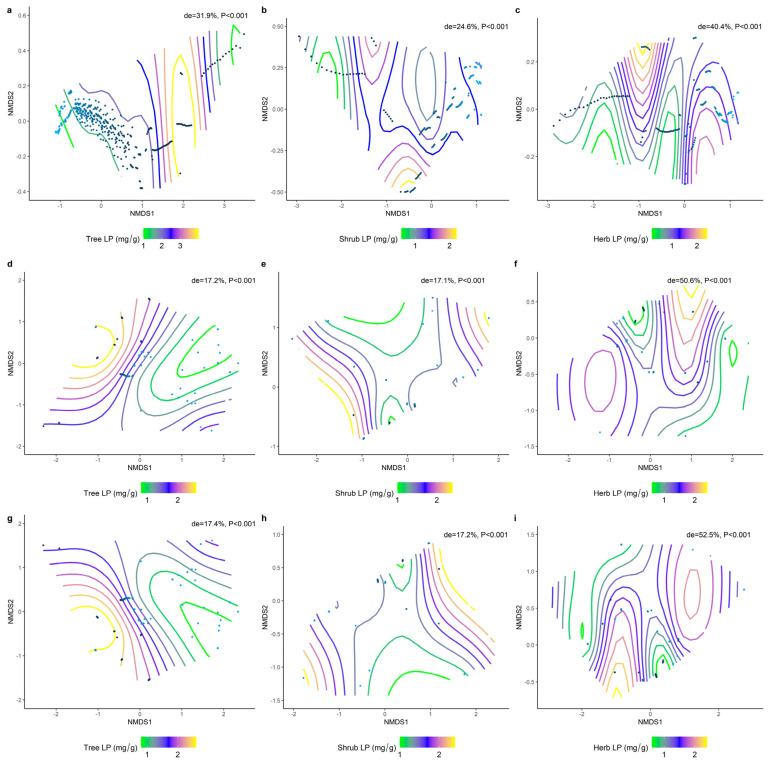
The NMDS ranking of climatic and soil factors with different life forms of LP. Value of de represents the deviation explained by the corresponding model. (**a**) NMDS ranking of soil factors with tree levels LP; (**b**) NMDS ranking of soil factors with shrub levels LP; (**c**) NMDS ranking of soil factors with herb levels LP; (**d**) NMDS ranking of climatic factors with tree levels LP; (**e**) NMDS ranking of climatic factors with shrub levels LP; (**f**) NMDS ranking of climate factors and herb levels LP; (**g**) NMDS ranking of the sum of soil factors and climate factors and tree levels LP; (**h**): NMDS ranking of the sum of soil factors and climate factors and shrub levels LP; (**i**): NMDS ranking of the sum of soil factors and climate factors and herb levels LP. Trait stacking indicates that abiotic factors, indicated by points on the NMD, are associated with higher or lower trait values, consistent with a colored trait gradient. Note that if the relationship between LP and abiotic factors is linear, the gradient splines will be parallel. Nonlinear relationships between LP and abiotic factors are represented by curve splines.

**Figure 11 plants-11-02171-f011:**
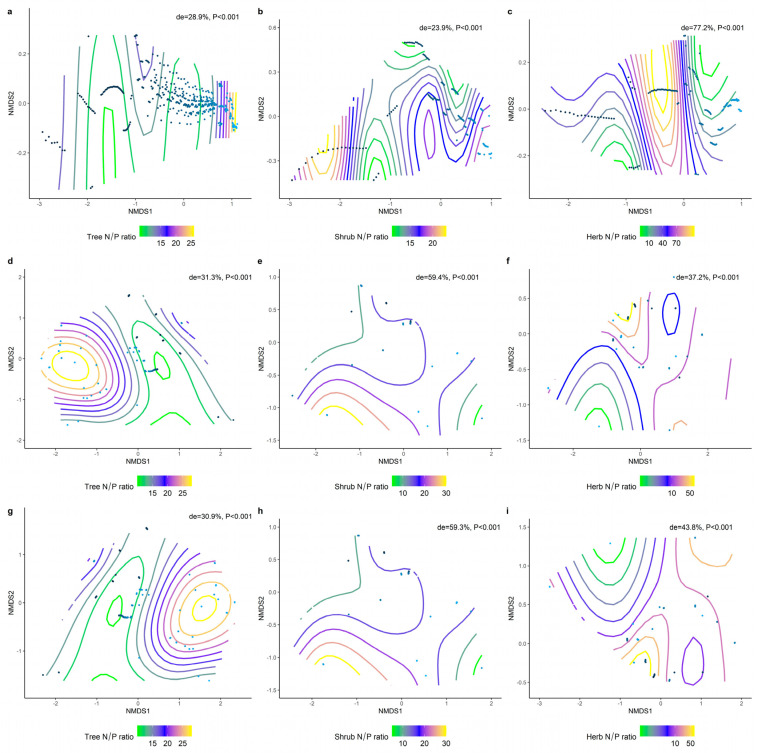
The NMDS ranking of climatic and soil factors with different life forms of N/P. Value of de represents the deviation explained by the corresponding model. (**a**) NMDS ranking of soil factors with tree levels N/P; (**b**) NMDS ranking of soil factors with shrub levels N/P; (**c**) NMDS ranking of soil factors with herb levels N/P; (**d**) NMDS ranking of climatic factors with tree levels N/P; (**e**) NMDS ranking of climatic factors with shrub levels N/P; (**f**) NMDS ranking of climate factors and herb levels N/P; (**g**) NMDS ranking of the sum of soil factors and climate factors and tree levels N/P; (**h**) NMDS ranking of the sum of soil factors and climate factors and shrub levels N/P; (**i**) NMDS ranking of the sum of soil factors and climate factors and herb levels N/P. Trait stacking indicates that abiotic factors, indicated by points on the NMD, are associated with higher or lower trait values, consistent with a colored trait gradient. Note that if the relationship between N/P and abiotic factors is linear, the gradient splines will be parallel. Nonlinear relationships between N/P and abiotic factors are represented by curve splines.

## Data Availability

Not applicable.

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
