# Peer review of "Climatic Factors Determine the Distribution Patterns of Leaf Nutrient Traits at Large Scales"

_plants, 2022, doi:10.3390/plants11162171_

Round 1

Reviewer 1 Report

Accept in present form

Author Response

Thanks for your affirmation of our article.

Reviewer 2 Report

The authors made a paper regarding the Climatic Factors Determine the Distribution Patterns of Leaf Nutrient Traits at Large Scales. Good and extensive Results part.  I suggest the following, in order enrich the manuscript and provide its best shape:

Please avoid personal manner of addressing "we" and "our". the text will sound more professional.

Text also must be moderate English revised, some sentences are annoying repetitive (i.e. L341and 342, repetitive "LP is determined ...") or have editing mistakes (e.g. L381. g( ) ?). Please revise)

Discussion section must be improved, it is much too short. Few ideas to be developed/better developed: What influence do both soil management and climate factors have regarding leaf nutrients traits? I suggest checking and referring tohttps://doi.org/10.1007/s11356-021-14127-7  

How do you explain the soil enzymology and fertilization related to nutrients accumulation? https://doi.org/10.37358/RC.18.10.6590 and  https://doi.org/10.37358/RC.19.10.7576 

Moreover, what is the future direction, influence, impact expected in using nanotechnology in fertilizers field to achieve better /optimized results related to nutrients accumulation (few ideas in https://doi.org/10.1016/j.chemosphere.2021.132533 )

After L317, in a separate last paragraph of Discussion, please provide the strengths and the weakness of this study.

L347. Formula/equation needs reference.

L348. where not Where, as it is in the same phrase with the above text.

Author Response

Dear Reviewer,

We really greatly appreciate you for processing our manuscript entitled “Climatic factors determine the distribution patterns of leaf nutrient traits at large scales” (Manuscript ID: plants-1812814). We are grateful for your valuable suggestions and comments on the manuscript. We have polished the whole manuscript based on the comments, in order to ensure that it is clear and as brief as possible, following the Plants format.

The point-by-point responses to your comments can be found below. The comments are shown in black font and the responses are shown in blue font.

Yours sincerely,

Jie Gao and coauthors

########### Point-to-Point Response Letter ###############

The authors made a paper regarding the Climatic Factors Determine the Distribution Patterns of Leaf Nutrient Traits at Large Scales. Good and extensive Results part. I suggest the following, in order enrich the manuscript and provide its best shape:

Please avoid personal manner of addressing "we" and "our". the text will sound more professional.

Response: We are extremely grateful to the reviewer for pointing out this problem. Based on your suggestion, we have revised the full text. For example: We modify this sentence " Based on field survey data from 89 sites in China, we sought to identify the key drivers influencing leaf nutrient traits at different life forms in forest communities at the macro scale." to this: " Based on field survey data from 89 sites in China, an attempt was made to identify the key drivers influencing leaf nutrient traits at different life forms in forest communities at the macro scale.". We made similar changes elsewhere.

Text also must be moderate English revised, some sentences are annoying repetitive (i.e. L341and 342, repetitive "LP is determined ...") or have editing mistakes (e.g. L381. g( ) ?). Please revise)

Response: Thank you very much for your reminder and suggestion. Here is where we mistakenly wrote the first LN measure as LP. Here we have done the following: " In this experiment, LN is determined using the national standard method Kjeldahl method, LP is determined using the vanadium-molybdenum yellow absorbance method, and N/P is equal to LN/LP." modification. We have corrected "g()" to "g(•)", and all the rest have been modified. In order to avoid the existence of similar errors, we have carefully checked the full text.

Discussion section must be improved, it is much too short. Few ideas to be developed/better developed: What influence do both soil management and climate factors have regarding leaf nutrients traits? I suggest checking and referring to https://doi.org/10.1007/s11356-021-14127-7

How do you explain the soil enzymology and fertilization related to nutrients accumulation? https://doi.org/10.37358/RC.18.10.6590 and https://doi.org/10.37358/RC.19.10.7576

Moreover, what is the future direction, influence, impact expected in using nanotechnology in fertilizers field to achieve better /optimized results related to nutrients accumulation (few ideas in https://doi.org/10.1016/j.chemosphere.2021.132533 )

Response: Thank you for providing us with these excellent references. Based on your suggestion, we have added a lot of content to the discussion section. Also, in the text we cite more on "Soil Enzymology and Soil Nutrients" based on the references you provided. These are on lines 158-160, 174-178 and 185-190 respectively.

After L317, in a separate last paragraph of Discussion, please provide the strengths and the weakness of this study.

Response: Thank you for your suggestion. The strengths of our research are a variety of analytical methods, a large amount of data, and a wide range of research. In addition, we quantified the relative contributions of different environmental factors in shaping leaf nutrient functional traits, indicative of understanding the impact of global climate change on plant physiology. The shortcoming of our study is that the study did not involve more biological factors. All the above are presented in the last paragraph of the Discussion section.

L347. Formula/equation needs reference.

Response: Thank you very much for your reminder. We've added references to formulas and equations. References are made to the formulas and equations in the explanation of the formulas and equations in the text.

L348. where not Where, as it is in the same phrase with the above text.

Response: Thank you for your comments. We have corrected "Where" to "where". In addition, we have also checked the full text to ensure that there are no similar errors.

**************************** END **************************

Reviewer 3 Report

This study explores the distribution patterns of leaf N and P contents (LN, LP) and N/P ratios (N/P) in communities and their controlling factors at large scale, which is novel. It’s an interesting topic, I enjoyed the reading and found the study is novel and timely, and provide insight to investigating the relative effects of Climate factors and soil nutrient factor in shaping the spatial variation patterns of LN and N/P . The method are reliable and the figures are beautiful. I cannot find any problem of this manuscript.

Author Response

Thanks for your praise of our MS.

Reviewer 4 Report

Authors investigated the distribution pattern and influencing factors of nutrient functional traits in leaves of communities. This study is backed by a wealth of data. The innovation of this study is that the authors focused on functional traits at the community level, which is a great progress in the field of functional traits. The method of this research is appropriate and reasonable, and the research results are very rich and reasonable, which is a very good research. Most importantly, it quantifies the major role of environmental factors in shaping the spatial distribution patterns of functional traits and helps us better understand the effects of global climate change on plants.  I agree to accept this article.

Author Response

Thanks for your praise of our article.

This manuscript is a resubmission of an earlier submission. The following is a list of the peer review reports and author responses from that submission.

Round 1

Reviewer 1 Report

Comments to the Author

In this manuscript, authors used a huge data to examine the distribution patterns of community-level leaf N and P contents (LN, LP) and N/P ratios (N/P) and their controlling factors in China. This work stressed that it was of great significance to understand the distribution of plant N and P nutrient traits from the perspective of different forest vertical layers. Despite the potential significance of the work, several issues need to be addressed in order to have greater confidence in this paper.

Major issues are below.

1.     In this paper, authors aim to examine the influence of climate and soil factors on these community-level leaf nutrient traits. However, authors fail to elaborate clearly some key issues. For example, why do researchers need to study the patterns of leaf nutrient traits at the community level? What are the differences between species and community levels? Which indicator would be adopted to represent the community-level data, and what is its ecological significance? Moreover, the progress in the research related to community-level leaf nutrient traits is lack in the section of Introduction. As far as I know, there have been many studies performed in this field.

2.     These hypotheses raised in this paper seem unreasonable, due to the lack of sufficient theoretical explanation and derivation. For the first hypothesis, authors fail to explain the difference in physiology or other aspects among different forest vertical layers, although the comparison of leaf nutrient traits among growth forms have been reported extensively. For the second and third hypotheses, the introduction section synthetize some works correlating leaf nutrient traits with climate/altitudinal gradients, but why climate, especially MAT, would have a dominant effect on reabsorption was not shown. These biological mechanisms or causal relationships between them are key aspects to improve our understanding of the variation in nutrient traits.

3.     Most of analyses and results only describe simply the patterns and their relationships with climate and soil factors, but are lack of in-depth data mining. For example, I beg to differ that the relative roles of each environmental factors in leaf traits could be obtained only through the univariate analysis. How did authors deal with the multicollinearity between these climate and soil factors? Furthermore, in my opinion, NMDS is not an appropriate method to examine the relationships between leaf nutrient traits and environmental factors. NMDS is usually used to examine how the constituent species — or the composition — changes from one community.

4.     There is no denying that this paper is based on a huge data, including 818 forests, 69 sites in China. However, too many figures are provided in this manuscript. You’d better select some important figures or tables as the main results given in manuscript, while others need to be put in the supplementary file. 

5.     Some expressions are improper, e.g., vertical levels, intraspecific struggle, leaf nutrient profiles.

Other specific comments are below.

Abstract:

This description in this section is unclear and need further refinement. Authors need to give the definition of different vertical levels, and explain the relationships between “vertical levels” and “community level”. Moreover, you’d better give some details to explain how did authors draw this conclusion “the effect of climate on leaf nutrients is higher than soil factor”.

2 Results

L85-97 These two paragraphs should be merged into one unit. Provide the figure or table number in the manuscript.

L99-l34 This section is too long, and some description is repetitive. Please refine it. 

L137-144 I I'm afraid I don't totally agree with that you can obtain these results through the NMDS method.

There are too many figures. Maximum of 8 figures and tables are suitable for research paper. You can put some of them into the supporting information.

Discussion

These sections are rather descriptive and also lack logical deduction or logical induction. Moreover, some sentences are repeating the results.

Material and methods

L220-223 Many importance information about the field sampling is missing, including the forest types, quadrat size and number per site, forest origin, the altitudinal range, the method to sample leaves ……

L226-234 Please provide the method to calculate the value of different forest vertical layers.

L239-248 Why did not you use the soil physical and chemical data obtained from field measurement?

L250-275 Data analysis used in this manuscript seems inadequate for your research purpose.

References:

There are too many references, about 50 references are recommended.

Author Response

Dear Reviewer1,

We really greatly appreciate you for reviewing our manuscript entitled “Climatic factors determine the distribution patterns of leaf nutrient traits at large scales” (Manuscript ID: plants-1812814). We are grateful for your valuable suggestions and comments on the manuscript. We have polished the whole manuscript based on the comments, in order to ensure that it is clear and as brief as possible, following the Plants format. We also thank Elizabeth Tokarz at Yale University for her assistance with English language and grammatical editing.

The point-by-point responses to your comments can be found below. The comments are shown in black font and the responses are shown in blue font.

Yours sincerely,

Jie Gao and coauthors

########### Point-to-Point Response Letter ###############

In this paper, authors aim to examine the influence of climate and soil factors on these community-level leaf nutrient traits. However, authors fail to elaborate clearly some key issues. For example, why do researchers need to study the patterns of leaf nutrient traits at the community level? What are the differences between species and community levels? Which indicator would be adopted to represent the community-level data, and what is its ecological significance? Moreover, the progress in the research related to community-level leaf nutrient traits is lack in the section of Introduction. As far as I know, there have been many studies performed in this field.

Response: Thanks for your valuable suggestions. We chose to conduct research at the community level because the species level ignores asymmetric competition due to the difference of abundance among different species.and there is a theoretical risk in predicting the functional traits of individual plants due to the influence of interspecific relationships (e.g., intraspecific competition, mutualism). This is already been mentioned in our Introduction and Methods sections. Thank you very much for your suggestion, but we don't think it is necessary to introduce the functional traits of the community in a separate paragraph, because it is well known by most experts who do functional traits. The Species Issue we founded in the journal Plants is mainly to attract readers in the direction of functional traits. We have previously published a large number of studies on functional traits of communities (Please refer to the references).

These hypotheses raised in this paper seem unreasonable, due to the lack of sufficient theoretical explanation and derivation. For the first hypothesis, authors fail to explain the difference in physiology or other aspects among different forest vertical layers, although the comparison of leaf nutrient traits among growth forms have been reported extensively. For the second and third hypotheses, the introduction section synthetize some works correlating leaf nutrient traits with climate/altitudinal gradients, but why climate, especially MAT, would have a dominant effect on reabsorption was not shown. These biological mechanisms or causal relationships between them are key aspects to improve our understanding of the variation in nutrient traits.

Response: We are grateful for your suggestions. All our assumptions are based on the analysis made in the introduction. According to your suggestion, we found many deficiencies and revised the content of the introduction section to improve the rationality of the hypothesis. In order to avoid readers' doubts about the concept of vertical level, we modify the vertical layer to life form. We added the basis for the division of different life forms of plants, and also added some comparative frontier results on MAT as the dominant factor shaping leaf nutrient traits. For example,

Numerous previous studies have shown that temperature is a key factor determining changes in leaf nutrient traits, especially mean annual temperature (MAT) [3,11,12].

Forest communities are divided into different functional groups (e.g. trees, shrubs, and herbs) with distinct species composition and structure, and vertical differentiation provides a good indication of the community's adaptation to environmental conditions [22]

Most of analyses and results only describe simply the patterns and their relationships with climate and soil factors, but are lack of in-depth data mining. For example, I beg to differ that the relative roles of each environmental factors in leaf traits could be obtained only through the univariate analysis. How did authors deal with the multicollinearity between these climate and soil factors? Furthermore, in my opinion, NMDS is not an appropriate method to examine the relationships between leaf nutrient traits and environmental factors. NMDS is usually used to examine how the constituent species - or the composition - changes from one community.

Response: Thank you for your suggestion. We apologize that some of our descriptions were inappropriate and misled you. Effects of climate and soil on functional traits were not only calculated by NMDS but obtained using the Generalized Additive Model (GAM) and then mapped onto NMDS (Please refer to this reference: https://doi.org/10.1111/nph.16976). This has been supplemented and improved in the methods section. In addition, we have also considered the multicollinearity as you said, GAM is specifically designed to account for the spatial relationships of multiple factors. Fewer indicators can well avoid the correlations between factors, which is why we have chosen fewer indicators. Our method is feasible, and many excellent articles use the method, such as this one (https://doi.org/10.1111/nph.16976).

There is no denying that this paper is based on a huge data, including 818 forests, 69 sites in China. However, too many figures are provided in this manuscript. You’d better select some important figures or tables as the main results given in manuscript, while others need to be put in the supplementary file.

Response: Thank you for your suggestion. The scope of our research is large, involving different life forms, and different biotic and abiotic factors, so the results are numerous and all very important. Nevertheless, we have followed the advice you gave and put some of the important pictures in the accompanying figures. We have also carefully checked the entire paper and word count and revised and refined it.

Some expressions are improper, e.g., vertical levels, intraspecific struggle, leaf nutrient profiles.

Response: Thank you very much for your reminder. We have revised these inappropriate expressions. We check and revise the whole paper..

Abstract:

This description in this section is unclear and need further refinement. Authors need to give the definition of different vertical levels, and explain the relationships between “vertical levels” and “community level”. Moreover, you’d better give some details to explain how did authors draw this conclusion “the effect of climate on leaf nutrients is higher than soil factor”.

Response: Thank you for your comments. We have revised this part of the deficiencies to improve the deficiencies in our description of the results

2 Results

L85-97 These two paragraphs should be merged into one unit. Provide the figure or table number in the manuscript.

L99-l34 This section is too long, and some description is repetitive. Please refine it.

L137-144 I I'm afraid I don't totally agree with that you can obtain these results through the NMDS method.

There are too many figures. Maximum of 8 figures and tables are suitable for research paper. You can put some of them into the supporting information.

Response: We are extremely grateful to the reviewer for pointing out this problem. In the results section, we have all deleted it with reference to your suggestion. Duplicates were removed without changing the overall experimental results. The NMDS method, and the feasibility of GAM, we have explained earlier.

Discussion

These sections are rather descriptive and also lack logical deduction or logical induction. Moreover, some sentences are repeating the results.

Response: Thank you very much for your reminder and suggestion. We re-write the section of discussion.

Material and methods

L220-223 Many importance information about the field sampling is missing, including the forest types, quadrat size and number per site, forest origin, the altitudinal range, the method to sample leaves ……

Response: Thank you very much for your suggestions. Our study sites ranged from 19.1° to 53.5°N and 79.7° to 129.3°E. Forests at different sites were our research objects (Fig. S1). Forests included natural forests (almost covering all forest types in China) and plantations. About 4 adjacent plots (20 m × 20 m) were randomly selected in each forest. Locations (e.g., Latitude, longitude and elevation) of each plot were investigated. All trees in the plot were identified using their scientific name and confirmed using authentic herbarium specimens housed in different regions of China. Then every forest plot was divided into 16 subplots (5 m × 5 m). One 5 m × 5 m sample shall be set at the four corners 1 m from the edge of the sample plot. We investigated the species, number, height and coverage of shrubs. We selected 4 small samples of 1 m × 1 m from 4 corners of each 5 m × 5 m large sample to investigate the species, plant height and coverage of herbs.The specific methods to sample leaves can refer to [50].

L226-234 Please provide the method to calculate the value of different forest vertical layers.

Response: Thank you very much for your suggestions. We changed vertical layers to life forms.

L239-248 Why did not you use the soil physical and chemical data obtained from field measurement?

Response: Thank you very much for your suggestions.The physical and chemical properties of soil in small areas have strong heterogeneity, which is difficult to represent the overall situation of a region. Because the area we study is large, we choose large-scale soil data.

References:

There are too many references, about 50 references are recommended.

Response: Thank you very much for your reminder. A large number of references we do cite and some relatively insignificant references have been omitted based on your suggestion. Currently, we keep only 49 references.

In view of your excellent opinions, we welcome you to contribute to our Species Issue founded in the journal Plants (https://www.mdpi.com/journal/plants/special_issues/Morphology_Geometry).

References

  1. Gao J, Song ZP, Liu YH. Response mechanisms of leaf nutrients of endangered plant (Acer catalpifolium) to environmental factors varied at different growth stages. Global ecology and conservation,2019, DOI:10.1016/j.gecco.2019.e00521.
  2. Gao J, Wang KQ, Zhang X. Patterns and drivers of community specific leaf area in China.Global ecology and conservation. 2021.
  3. Gong HD, Zhang X, He X, Gao J*, et al. Latitudinal and climate effects onkey plant traits in Chinese forest ecosystems. Global ecology and conservation, 2019, DOI:10.1016/j.gecco.2019.e00527
  4. Gong HD, Yao FG, Gao J*. Succession of a broad-leaved Korean pine mixed forest: functional plant trait composition.Global ecology and conservation,2020, DOI: 10.1016/j.gecco.2020.e00950

**************************** END **************************

Reviewer 2 Report

few technical corrections:

Line 85 and in all manuscript

Correct 2.1. . Distribution……… in 2.1. Distribution ….

in Figure 2,3, 4, 5, 6, 7 and 8 legends add the description of bars or lines 

Red –tree

Green herb

Blue- shrub

Author Response

Dear Reviewer,

We really greatly appreciate you for processing our manuscript entitled “Climatic factors determine the distribution patterns of leaf nutrient traits at large scales” (Manuscript ID: plants-1812814). We are grateful for your valuable suggestions and comments on the manuscript. We have polished the whole manuscript based on the comments, in order to ensure that it is clear and as brief as possible, following the Plants format.

The point-by-point responses to your comments can be found below. The comments are shown in black font and the responses are shown in blue font.

Yours sincerely,

Jie Gao and coauthors

########### Point-to-Point Response Letter ###############

Comments and Suggestions for Authors

few technical corrections:

Line 85 and in all manuscript

Correct 2.1. . Distribution……… in 2.1. Distribution ….

in Figure 2,3, 4, 5, 6, 7 and 8 legends add the description of bars or lines

Red –tree

Green- herb

Blue- shrub

Response: Thank you for your corrections to some of the inadequacies of our article. Based on your instructions, we have revised and improved the article. We also thank Elizabeth Tokarz at Yale University for her assistance with English language and grammatical editing.

**************************** END **************************

Reviewer 3 Report

The authors made a paper regarding the Climatic Factors Determine the Distribution Patterns of Leaf Nutrient Traits at Large Scales. 

Good and extensive Results part.  I suggest the following, in order to enrich the manuscript and provide its best shape:

Shape suggestions

Please check the instructions for authors regarding the figures, which must be inserted as closest is possible to their mention in the main manuscript (not at the final of the manuscript).

Also check the order of the sections.

Revise English please, avoiding the personal manner of addressing (we and our), and using the impersonal one. The text will sound much more professional.

Conclusions. L215. The statement “This study has important implications for our understanding of…” is not the best expression. Please reshape it as This study has important implications for a better understanding of ….” (not for your understanding)

Content suggestions

Discussion section must be improved. Few ideas to be developed: What influence do both soil management and climate factors have regarding leaf nutrients traits? I suggest checking and referring to https://doi.org/10.1007/s11356-021-14127-7  

How do you explain the soil enzymology and fertilization related to nutrients accumulation? https://doi.org/10.37358/RC.18.10.6590 and  https://doi.org/10.37358/RC.19.10.7576 

Moreover, what is the future direction, influence, impact expected in using nanotechnology in fertilisers field to achieve better /optimised results related to nutrients accumulation (few ideas in https://doi.org/10.1016/j.chemosphere.2021.132533 )

In a separate last paragraph of Discussion, after L205, please provide the strengths and the weakness of this study.

Author Response

Dear Reviewer,

We really greatly appreciate you for processing our manuscript entitled “Climatic factors determine the distribution patterns of leaf nutrient traits at large scales” (Manuscript ID: plants-1812814). We are grateful for your valuable suggestions and comments on the manuscript. We have polished the whole manuscript based on the comments, in order to ensure that it is clear and as brief as possible, following the Plants format. We also thank Elizabeth Tokarz at Yale University for her assistance with English language and grammatical editing.

The point-by-point responses to your comments can be found below. The comments are shown in black font and the responses are shown in blue font.

Yours sincerely,

Jie Gao and coauthors

########### Point-to-Point Response Letter ###############

Shape suggestions

Please check the instructions for authors regarding the figures, which must be inserted as closest is possible to their mention in the main manuscript (not at the final of the manuscript).

Also check the order of the sections.

Revise English please, avoiding the personal manner of addressing (we and our), and using the impersonal one. The text will sound much more professional.

Conclusions. L215. The statement “This study has important implications for our understanding of…” is not the best expression. Please reshape it as “This study has important implications for a better understanding of ….” (not for your understanding)

Response: Thank you very much for your suggestion. Firstly, the order of the article is strictly based on the format of Plants, so we apologize for the inconvenience of reading it. Secondly, we have revised the use of personal pronouns in response to your reminder.

Content suggestions

Discussion section must be improved. Few ideas to be developed: What influence do both soil management and climate factors have regarding leaf nutrients traits? I suggest checking and referring to https://doi.org/10.1007/s11356-021-14127-7  

How do you explain the soil enzymology and fertilization related to nutrients accumulation? https://doi.org/10.37358/RC.18.10.6590 and  https://doi.org/10.37358/RC.19.10.7576

Moreover, what is the future direction, influence, impact expected in using nanotechnology in fertilisers field to achieve better /optimised results related to nutrients accumulation (few ideas in https://doi.org/10.1016/j.chemosphere.2021.132533 )

In a separate last paragraph of Discussion, after L205, please provide the strengths and the weakness of this study.

Response: In the discussion section, we have made an overall revision to improve the logic of the content and add some missing theories. As for the resources you provide, we have done careful research and refined our presentation based on the information you provided. But what needs to be explained to you is that what we want to express is: the activity of soil microorganisms to decompose soil nutrients, not soil enzymes. Here is a misunderstanding caused by our improper expression. Finally, based on your suggestion, we have added the strengths and weaknesses of this paper at the end of the discussion, and we also look forward to furthering research in the future.

**************************** END **************************

Round 2

Reviewer 1 Report

Compared with the first submission, this version has witnessed some improvement in scientific writing. But some questions needed to be further answered.

(1)    Although these results of temperature acting as a key factor determining changes were given in this revised version, I did not think that these research examples can infer directly that MAT is the dominant factor affecting leaf nutrient traits among so many climatic and soil factors.

(2)    L72-74 Form these sentences, authors seems to want to emphasize the importance of community-level analysis of leaf nutrient traits. But, in the Authors' Responses to Reviewer's Comments, authors stated that “it is necessary to introduce the functional traits of the community in a separate paragraph, because it is well known by most experts who do functional traits” “We have previously published a large number of studies on functional traits of communities”. Therefore, I don't know what is the focus of your research.

(3)    Also, if the community-level traits are not the subject of this manuscript, why did you give the maps of CWM values in the Result section? This result seems unrelated to your topical subject (difference in plant functional groups) ?

(4)     L72-74 As authors stated, “ comparisons of the leaf nutrient traits of different plant forms are mostly conducted at the species scale, few studies have been conducted at a larger spatial scale or at the community scale” , then your study is conducted on species level or community level ? I cannot find out how authors obtained or calculated the value of different forest vertical layers. If authors obtained the values of different forest vertical layers through averaging the values within the same growth forms, in my opinion, such analysis is still conducted on the species level.

(5)    Although authors explained the rationality of methods used in this manuscript, the results are hard to understand. For example, which figures or tables can make us draw this conclusion “Climatic factors played a greater role in shaping the spatial variation in LN and N/P than did soil nutrient factors” ? I think some widely-used methods, e.g., stepwise regression, should be selected to qualify the effects of environmental factors on leaf traits.

Author Response

Dear Reviewer1,
We really greatly appreciate you for reviewing our manuscript entitled “Climatic
factors determine the distribution patterns of leaf nutrient traits at large scales”
(Manuscript ID: plants-1812814) again. Please see attachment for detailed reply.
